# Interest Differences and Organizational Learning

**Laurie Field**

Faculty of Human Sciences, Macquarie University, Sydney, NSW 2109, Australia;
laurie.field@mq.edu.au; Tel.: +61-298-508-683

**Abstract:** This paper argues that interest differences are the key to understanding the nature of organizational learning and the processes by which it occurs, yet the concept of 'interest' is very much underdeveloped in the organizational learning literature. Drawing on the work of Habermas and Lukes, the paper proposes a model of the relationship between shared learning and interests and elaborates on it using a case study of pay and performance management change at a large Australian finance-sector company, DollarCo. The case study provides many examples of shared learning associated with both common and competing interests, including a great deal of learning resulting from tensions between DollarCo's economic and technical interests, on the one hand, and employees' ontological interests on the other. By doing so, it underlines the value of foregrounding interests and interest differences in studies of workplace and organizational learning and raises questions about the extent to which many published accounts of so-called 'organizational' learning are actually describing 'shared interest group' learning.

**Keywords:** interests; organizational learning; power; workplace learning

## 1. Introduction

In recent decades, many investigators have looked at the nature of organizational learning and the processes by which it occurs. This paper argues that in such investigations, interests and interest differences deserve far more attention than they have received to date. It proposes a model of interests which provides a basis for thinking about the relationship between interests and shared learning, and for considering the circumstances under which shared learning becomes 'organizational'. Data from a large Australian finance-sector company, DollarCo, where the pay and performance management systems were changing, are then used to elaborate on this model. The argument running through the paper is that foregrounding 'interests' opens the way for a richer conception of learning at work, one which helps to clarify the circumstances under which shared learning becomes 'organizational'.

When one looks at the literature of organizational learning, the concept of 'interest' is very much underdeveloped. If the term is used at all in the organizational learning literature, it tends to mean 'curiosity about'. Thus, one finds many references to such things as "interesting finding...researcher interests...interested respondents...interesting patterns...management interest in 'organizational' learning".

Even when mention of 'interest' extends beyond 'curiosity about' to refer to 'common interest', there is a tendency in the organizational learning literature to treat 'interest' and 'interest differences' as inconsequential and not worthy of elaboration. For example, Trong Tuan (2013) suggests that 'to activate the transformation of individual knowledge into organizational knowledge, managers should...inspire members to...transcend their self-interests [and] develop organizationally beneficial behaviors', and later contrasts the alternatives of 'organizational members indulging their own interests' with 'contribut[ing] to the success of the [whole] organization' (Trong Tuan 2013, both p. 219), without elaborating on the self-interest/organizational-interest dichotomy and without considering how it relates to organizational learning. In another discussion that touches on interests in the context of workplace learning, Ordóñez de Pablos (2005) explains that 'society can be viewed as a market where

individuals exchange diverse ideas and goods in pursuit of interest' and that 'the interest [sic] of some is better served than the interests of others' (p. 438). However, as with Tuan's account, there is no elaboration on the relationship between interests and learning.

Treating interest differences as unworthy of close consideration is part of the general tendency in studies of organizational learning to gloss over politics and power (Huzzard 2004). To an extent, this reflects the North American origins of much of the foundational organizational learning literature and an assumption that the interests of managers and employees can be reconciled through human resource management strategies such as formal and psycho-cultural contracts. In contrast, in countries like Australia, the UK and Canada, which have strong industrial relations traditions, it tends to be assumed that the interests of employees and employers are always in conflict and can only be managed through collective bargaining (Newman and Newman 2015). This more pluralistic perspective raises questions about the likelihood that all organizational members will experience the degree of security and organizational alignment necessary to share their learning in service of their organizations.

Admittedly, some accounts of knowledge management and workplace learning have considered interest differences. For example, in their discussion of knowledge management, Kimmerle et al. (2008) underline the importance of context in determining whether knowledge serves organizational interests. To illustrate, they discuss the example of managers engaged in contract negotiations who, while doing so, clearly represent organizational interests; but afterwards, the same individuals in a different context (e.g., private socializing) may serve their own or some other interests. In studies of computer-mediated knowledge communication via shared data-bases (e.g., (Olson and Olson 2003; Kollock 1998)), some consideration has also been given to the tension between individual and organizational interests. Given that there are costs of sharing knowledge (e.g., time involved for managers or employees) and benefits of withholding (e.g., maintaining status), individual interests may be best served by NOT sharing. But the situation is unlikely to be clear-cut because, at the same time, contributing to a shared knowledge pool may benefit individuals (Cress and Martin 2006). This tension, termed the 'information-exchange dilemma', has been examined in the knowledge management literature, with efforts to identify the antecedents (such as 'trust in management' and a 'conscientious stance towards one's organization') of preparedness to share individual knowledge (e.g., (Fang and Chiu 2010)).

The workplace learning field also contains studies which consider the role of shared and competing interests, knowledge and learning (e.g., (Bjerg Hall-Andersen and Broberg 2014; Järvensivu and Koski 2012; Barrett and Oborn 2010; Fenwick 2008; Contu and Willmott 2003; Forrester 2002)). For example, Bjerg Hall-Andersen and Broberg (2014) draw on case study data to consider learning across knowledge boundaries in organizational settings. They show that learning at work tends to reside at the level of individuals and of 'pockets' associated with particular domain-specific conditions relating to management, power, pre-existing practices and resource considerations. A number of investigations (including Barrett and Oborn (2010) and Swan et al. (2007)) have considered the role that particular objects (and 'thought leaders' skilled at attributing particular meanings to these objects) play in generating commitment to sharing, assessing and applying knowledge in the face of different interests and world-views.

Although studies like these, from the fields of knowledge management and workplace learning, acknowledge interest differences, in most cases reference to 'interests' only occurs indirectly. For example, in their investigation of lean production training in the manufacturing sector, Yasukawa et al. (2014) barely mention 'interests', but nevertheless note the contested nature of knowledge and learning during workplace change, highlighting tensions between learning programs designed to increase productivity and workers' concerns about protecting their jobs and salaries. Similar themes provide the basis for Järvensivu and Koski (2012) account of employees' opposition to learning associated with work intensification. Examples of indirect attention to 'interests' can also be found in the extensive 'communities of practice' literature (Lave and Wegner 1991; Wenger 2000). Certainly, common interest provides the focus of knowledge and learning in communities of practice studies, but the concept of 'interest' is secondary to consideration of knowledge transfer and learning processes.

As these examples illustrate, while some scholars of knowledge management and workplace learning have considered the role of interests, the depth of treatment has tended to be limited. Moreover, in the field of organizational learning, which is the main focus of this paper, very little attention has been paid to interests and interest differences, even in accounts which consider the political and emotional dimensions of learning (e.g., (Vince 2001)). This tendency has important consequences for understanding the nature of organizational learning and the processes by which it occurs. To illustrate, consider an organization in which a researcher is studying a workplace problem which impacts on the interests of various organizational actors. Imagine that the focus of the investigation is on organizational learning. Without considering interest differences, as organizational members attempt to resolve the problem, our researcher might report many examples of learning and might claim that these constitute 'organizational' learning.

The implicit assumption is that the organization constitutes, or at least should constitute, a single shared interest group aligned around common economic and technical interests, and that these provide the locus for organizational learning. Unitary assumptions—in this case, a view of organization as single shared interest group, reflected in such things as shared mental models (Kim 2004) and collective competence (Ohlsson 2014) that accord with the organization's strategic priorities—underlie this conception of what happens.

From a unitary perspective, divergence from the (assumed) common interest in addressing and resolving workplace challenges, and learning in the process, tends to be viewed negatively. This tendency can be traced back to the earliest significant writing about organizational learning. For example, Argyris (1994) depicts learning as something happening in a rational 'computer' part of oneself. In this conception, defensive routines associated with protecting against feelings of vulnerability and embarrassment (or what, in the terminology of this paper, would be termed 'defensiveness associated with the ontological interest') are viewed as 'anti-learning' (p. 79) and 'a recipe for ineffective learning' (p. 80). In his seminal account of organizational learning, Senge (1990) describes organizational politics as a 'reeking odor' and 'a perversion of truth and honesty' (p. 273) that organizational learning (and particularly his unitary depiction of learning) has the potential to counter. Similar thinking is found throughout much of the organizational learning literature, particularly in material originating in North America. Reluctance to engage in learning which serves the interests of the organization has been referred to variously as 'learning obstacles' (McGill and Slocum 1993), 'learned helplessness', (Marsick and Watkins 1994) and 'knowledge-inhibiting activities' (Leonard-Barton 1995). In similar vein, Snyder and Cummings (1998) insist that shared learning at work is 'healthy' and, after considering impediments to achieving this 'healthy' outcome, explain how to 'diagnose' what they refer to as 'learning disorders' such as 'self-interest' and 'powerlessness'. Costley (2001) refers to the 'problem of conflict of interest' (p. 62) between individuals' learning requirements and an organization's focus on business goals, but does not elaborate on 'conflict of interest'. More recently, Godkin and Allcorn (2009) offer a diagnosis of behavioral barriers to organizational learning, with 'narcissism' and what they term 'arrogant organization disorder' considered particularly problematic; and Smith (2011) refers to the 'catastrophic' (p. 6) impact of being governed by 'self-interest', again without elaboration.

## 2. The Benefits of Foregrounding Interests in Organizational Learning Studies

There are two main reasons why interests and interest differences should be considered closely in organizational learning studies. First, studies of organizational knowledge and workplace learning which acknowledge interest differences suggest that foregrounding interests is likely to illuminate the nature of organizational learning. Consider, for example, Thursfield's (2008) account of efforts to promote collective learning within a UK local authority, efforts which had limited success; and the account by Järvensivu and Koski (2012) of six workplaces where the employer tried to facilitate workplace learning. While the concept of 'interest' is not considered in depth in either paper, at least these accounts do take interest-differences into account, with Thursfield referring to 'self-interest', 'personal interest' (including those of managers) and to the 'distributed interests' of organizational

actors, and Järvensivu & Koski to 'different interests' and 'interest-laden struggles'. The result is that, in place of the type of politically naïve pathologizing discussed in the last section and exemplified by Snyder and Cummings (1998) and Godkin and Allcorn (2009), both studies product a richer, more nuanced picture of shared learning, one which raises questions about the extent to which this learning is 'organizational'.

Second, there is ample evidence, associated with areas of scholarship such as the sociology of work, employee relations and critical accounts of workplace change (e.g., (Brown 2008; Sennett 2006; Clegg et al. 2006; Fleming 2005; Thomas and Davies 2005; Bain and Taylor 2000; Ackroyd and Taylor 1999)) of tensions between employers' economic and technical interests and employees' interest in retaining jobs and maximizing income, status and resources. In fields of literature such as these, conflict is assumed to be structurally inherent in the employment relationship, reflected by interest groups pursuing their (to some extent) dissonant agendas against a backdrop of asymmetric power relations and assumed managerial prerogative.

## 3. Perspectives on Interests and Learning

This section considers existing scholarship relating to the relationship between interests and learning. This relationship has been extensively studied in psychology, but the primary focus has been on individual interests and their relationship with text-based learning (see Silvia (2006)) for a detailed review). One finds a much better base for considering interests in the context of organizational learning in the literatures of work sociology and employee relations. The work of Habermas and Lukes are particularly helpful, and each is considered below.

While couched in abstract terms with no specific mention of organizational learning, Habermas's analysis of knowledge and interests provides insights into the ways in which learning and interests are linked. Habermas (1987) argues that the basic conceptual structures of human knowledge (and, by implication, learning) are shaped by interests that are deeply anchored in people's social existence. Habermas defines interests as 'the basic orientations rooted in specific fundamental conditions of the possible reproduction and self-constitution of the human species, namely *work* and *interaction*' (p. 196, italics in original). Resonant with this conceptualization, the kinds of interests considered in this paper are not about individual preferences or the gratification of basic day-to-day needs (e.g., 'an interest in studying organizations'), but instead relate to much more significant issues and challenges. From Habermas's perspective, these extend beyond the economic and technical realm, and encompass efforts to resolve challenges relating to achieving mutual understanding (which he terms the 'practical' interest) and freedom from political constraint (the 'emancipatory' interest).

Importantly, Habermas's work suggests that a great deal of learning results from tensions and incompatibilities between, on the one hand, the 'system' (institutions and financial markets and their instrumental rationality) and, on the other hand, human interest associated with the 'life-world' (the inter-subjective world of human experience and social action) (Brunkhorst 1999). This perspective is particularly noteworthy because it contrasts so markedly with the strong emphasis in the organizational learning literature on learning generated by alignment and compatibility with commercial interests rather than from 'tensions and incompatibilities' with commercial interests.

Habermas's commentary on knowledge and interests suggest that, in work settings, organizational members may be learning far more than the organizational learning literature tends to acknowledge. It also suggests that, when seeking to account for learning at work, interests and interest differences are worthy of close attention. Habermas's theorizing suggests that to account for learning in work settings, two different levels of interest need to be considered. First, learning is likely to be associated directly with the economic and technical interests of the organization. Learning of this type has formed the basis for most accounts of organizational learning to date.

Second, Habermas's analysis is suggestive of a deeper level of interests, one likely to be associated with a great deal of learning. Habermas's work suggests that, at this deeper level, there are two learning components, namely learning associated with organizational members' interest in (a) coming

together and achieving mutual understanding in the face of oppositional efforts by those associated with money and power; and (b) achieving freedom from political constraint in the face of attempts to have constraint imposed.

Lukes (2005) analysis of power, exercised in contexts where interests differ, provides a second framework for thinking about the relationship between interests and learning. Like Habermas, Lukes' focus is on wider society but, as with Habermas, it is not difficult to see implications for organizational learning. According to Lukes, in order to resolve interest differences, power may be utilized by management in three ways. The first, and most overt, way is the power to prevail over others, which Lukes terms the 'first face of power'. Consider the example discussed earlier, of a problem in a work setting, efforts to resolve it and resultant learning. If, in this situation, one were to study the processes of consultation and debate involved, and how these ultimately lead to problem resolution and solution implementation, then, in part, one would be considering Lukes' first face of power. This phrase refers to power associated with overt decision-making processes used to resolve interest differences and reach mutually satisfactory outcomes.

However, a close analysis of this situation might also reveal less visible, parallel processes occurring. Perhaps, as well as openly stating and justifying their position, management might attempt to control the problem-resolution agenda while ignoring, misrepresenting or deflecting the grievances of those whose interests are negatively impacted. Lukes calls the power to exclude threatening issues from debate and to shore up existing power arrangements the 'second face of power'.

Looking deeper still at this situation, one might find evidence of yet another level of power-based dynamics, this one only partly visible to actors and observers. Management in our example might exercise power to influence people's perceptions of options and of what is acceptable. For example, management might seek to induce employees to endorse arrangements despite the likelihood (for employees) of negative consequences; or they might try to steer away from solutions which, if it were not for management preference-shaping, employees would recognize as aligning with their real interests. Lukes terms the power to do so the 'third face of power'.

As with Habermas, Lukes' theoretical work is suggestive of two levels of interest and associated learning in situations like the one described in our example. At the more overt level, we would expect a great deal of shared learning to be directly associated with the economic and/or technical interests of the organization. The 'economic' interest might be reflected in the wish of all organizational members to ensure that the organization stays profitable, so that both managers and employees maximize their incomes and keep their jobs. At the same time, the 'technical' interest might be reflected in a shared commitment, by people responsible for part of an organization's operations, to resolve technical challenges. In commercial organizations, the economic and technical interests often intersect. For example, the learning of a group of financial analysts in a bank treasury office trying to accommodate a regulatory change with minimum system downtime may be responding to both an 'economic' interest that encompasses individual bonuses, team rewards and bank profitability, as well as to a 'technical' interest expressed by the phrase 'professional pride'. In pursuing their common 'economic' and 'technical' interests (to use Habermas's terms), groups of people in a workplace may learn a great deal, drawing on and supplementing the previously accumulated pool of learning about problem resolution.

As with Habermas's framework, Lukes' analysis suggests a second, covert level of interests and learning. We might expect learning by both managers and employees at this second level. For managers, learning at the second level is likely to relate to exercising the second and third faces of power—that is, to learning how to manipulate agendas (Lukes' second face of power) and to covertly shape preferences (Lukes' third face of power). At the same time, Habermas's analysis reminds us of what is likely to be happening on the employee side. Employees are likely to be learning to understand manifestations of the second and third faces of power so they minimize the likelihood of being duped by management.

### 4. A Model of Shared Learning and Interests

To summarize the last section, when one looks at organizational learning through a lens that foregrounds interests and interest-differences, one can differentiate between three types of shared learning (see Table 1). First, learning may be associated with the economic and technical interests of the organization. This type of learning forms the basis of most scholarship surrounding organizational learning.

Second, managers, both individually and in groups, are likely to be learning about the exercise of power, motivated by their own economic and technical interests and those of their organization. Third, employees, individually and in groups, are likely to be learning as a result of their ontological interest in protecting against such things as loss of status, loss of freedom, powerlessness, shame and loss of resources (for a fuller explanation of 'ontological interest, see Field (2012)). In pursuing the ontological interest, the achievement of mutual understanding that Habermas refers to plays a central role. The covert level of shared learning in organizations—learning to exercise power, and learning to protect against the exercise of power, see Table 1—has received little attention in the organizational learning literature to date. The next section looks briefly at why this may be so, and then the remainder of the paper considers interests and learning at the covert level. In doing so, it draws on a study of pay and performance management system change within a large Australian company, referred to here as DollarCo. The paper concludes by reflecting on the conditions under which the covert-level learning which occurred at DollarCo could be considered 'organizational'.

**Table 1.** A model of the relationship between shared learning and interests.

|  | | Focus of Learning | Primary Interests Served | Who Learns? |
|---|---|---|---|---|
| **Overt level** | 1. | Learning to resolve economic and technical challenges | Economic and technical interests | Both managers and employees |
| **Covert level** | 2. | Learning to exercise power | Economic and technical interests | Managers |
|  | 3. | Learning to protect against the exercise of power | Ontological interest | Employees |

### 5. Challenges of Including Interest Differences in Studies of Shared Learning at Work

There can be little doubt that part of the reason why interests and interest differences have received so little attention in the organizational learning literature is the dominance of unitary thinking, the unquestioning assumption that the interests of organizational members are, or at least should be, aligned. This is a theme that the author has addressed in detail elsewhere (see Field (2011)) and will not be repeated here.

However, there is second important reason for the omission, one which has not been discussed previously in the context of the organizational learning literature—namely, that in empirical investigations, 'interests' and 'interest differences' are difficult concepts to operationalize. Habermas's writing about interests is an abstract, general account of social existence, and the 'practical' and 'emancipatory' interest to which he refers may influence things without actors being aware of them or able to articulate the role that they play in shaping learning. Similarly, Lukes' second face of power may be exercised behind closed doors, out of view of researchers; and the third face may be hidden from those who exercise it as well as from those subjected to it.

The abstractness of Habermas's conception, and the covert nature of Lukes' second and third faces of power, represent real challenges for researchers seeking to operationalize 'interests' using the work of these authors. Moreover, it is evident from empirical work associated with the bodies of scholarship referred to earlier—the sociology of work, employee relations and critical accounts of workplace change—that the reality of interest differences at work may be both subtle and complex. The interests of an individual at work in relation to a particular issue may, to an extent, range over all the levels shown in Table 1. Importantly, nothing in the conceptual or empirical work that underpins

this paper suggests that there are hard and fast dividing lines between different categories of interest or between the interests of different organizational members.

Looking to the literature, even studies which deal explicitly with 'interest' tend to skim over discussion of the nature of individual and shared interests. As Shapiro (2006) notes, 'class, gender, status, religion, race and countless other bases of human identification can generate interests that can plausibly be ascribed to people' (p. 145), and 'there is likely to be considerable disagreement over just what interests are at stake in a given situation, how—if at all—they are being compromised, and how they might bear on other interests that might also be threatened' (p. 152).

Nevertheless, despite these challenges, in my view, those seeking to account for organizational learning have a choice. They can either adopt a unitary perspective that downplays or ignores interest differences and limits one's ability to account for organizational learning. Alternatively, if they can tolerate a degree of uncertainty about the interests of organizational actors, there is potential to produce a richer, clearer account of organizational learning and the processes by which it occurs.

## 6. DollarCo Case Study

As a basis for better understanding the relationship between shared interests and learning, let us now consider empirical data derived from a case study of the introduction of a new pay and performance management system (henceforth referred to as 'New PPM') at a large Australian finance-sector company, DollarCo. It was decided to study learning during pay and performance management system change because it was anticipated that doing so would provide a window on organizational learning in politically charged employee relations contexts where interests differ.

### 6.1. Methodology

Data collection occurred in two phases, each lasting around two months. The first phase occurred before New PPM was introduced; and the second, six months later, as the new approach was being implemented. Sampling of interviewees was purposive, with the aim of gathering data from a wide range of people likely to have different perspectives on learning associated with pay and performance management change, including managers at different levels, employees, internal consultants and union delegates. In all, 25 interviews were conducted at DollarCo, each typically lasting an hour or more.

All interviews were recorded, and the transcribed material was analyzed with the help of qualitative data analysis software. This involved working paragraph by paragraph through each transcript, attempting to understand the relationship between interests and learning and the requirements for individual learning to become organizational. The result of these activities was a detailed picture of interests and their relationship with learning at DollarCo in the context of a changing pay and performance management system.

### 6.2. Site Background

During the decade before the case study, DollarCo had undergone extensive transformation, all happening in the context of general instability in the Australian finance sector associated with deregulation, increased competition, globalization and ever-improving communications systems. The result of these changes was gradually to move back-office staff from small centers into very large centers such as the Suburban Complex discussed here. This Complex, born out of this turbulent decade, consisted of a Customer Center and a Product Center.

The two Centers had very different histories. The Customer Center had relatively young and well-educated staff, flexible work arrangements and a highly respected senior manager. The result was that levels of trust and commitment were high.

In contrast, Product Center staff were older and more technically qualified than those in the Customer Center. They valued their status and the responsibility associated with their work. In the years before moving to the Suburban Complex, the Product Center had experienced considerable

turmoil, with new management, sudden changes of direction, attempts to fragment and deskill work, significant job losses and industrial strife. At the time the case study began, levels of resistance remained high. Most Product Center employees were on a traditional remuneration system in which they were paid according to how long they had held their positions. For some time, senior DollarCo management had been keen to get rid of job grades, and their efforts ultimately gave rise to a fundamentally different, competency-based approach to pay and performance management, referred to here as 'New PPM'.

During the case study, the central challenge for management was to convince Product Center staff to accept New PPM. Their reticence was understandable, because New PPM potentially would have negative impacts. The vote on introducing New PPM to the Product Center finally occurred mid-way through the ten-month case study period. After maneuvering by the staff union that resulted in unexpected delays, New PPM was finally endorsed. The second round of interviews occurred during the implementation period that followed.

## 7. Examples of Shared Learning Associated with Management's Exercise of Power

Consistent with the conceptual analysis reported earlier (particularly Lukes' framework) and summarized in Table 1, DollarCo management learnt a great deal about exercising power during the case study period As the data were analyzed, it became clear that management's learning about exercising power at DollarCo primarily fell into three categories, namely learning about pluralist aspects of work, learning to apply pressure to perform and learning to manage information to achieve sought-after outcomes. Examples are given below.

### 7.1. Management Learning about Pluralist Aspects of Work

A great deal of shared learning at DollarCo related to management learning how employees' interests often differed from their own, a reality which contrasted markedly with management's wish for unitary alignment. This hope was evident in repeated references to 'one organization'. For example, according to a senior manager: 'We are trying to say that although we perform different functions, at the end of the day we are part of one organization, DollarCo.'

Throughout the struggle to convince the Product Center to accept New PPM, a great deal of management learning related to the challenge of getting employees to pull together, despite evidence that employees' interests often differed from those of the company. One team leader recounted a lesson learnt during the period of change: 'We are all individuals, we all have our goals, be it low, high or medium. The high achievers are a minority, and the majority is probably middle to low, "just come to work and go home" stuff.'

Another difference in perception that resulted in considerable learning by management was the contrast between employees' focus on job security, and managers' focus on productivity and profit. Asked what those driving the implementation of New PPM had learnt, one manager replied 'We've learnt that employees often think that the current way of doing things is best. When you suggest otherwise, you appear to be criticizing them, because they have such a strong investment in today's systems and processes.' One can speculate about the nature of this 'strong investment'. While it partly relates to a technical interest, the data suggested that it also related to the ontological interest in preservation of the status quo and in protection against vulnerability.

In response to the difficulties that they were facing transferring New PPM from the Customer Relations Center to the Product Center, the managers involved were learning about interest differences across the organization. As one commented: 'In the case of the two centers [the Product Center and the Customer Relations Center], you're not dealing with the same sort of people. People view Product Center work differently.'

The case study illustrates how it is possible to learn a great deal in one organizational context (the Customer Relations Center), but then have difficulty applying it in a different context (the Product Center). One interviewee likened the situation to a competition between opposing teams:

'I think in business, it is just the same [as in sport]. If [one state] is doing something well, you go and ask the relevant unit in [another state] why, and it will be because "they are fudging their figures there". There is no acknowledgment that they might be doing it better.'

An internal consultant who had been involved across several DollarCo sites supported these views. He summarized what he and his colleagues had learnt from involvement in the New PPM project: 'One of the main barriers to [learning] is that the organization doesn't work as one. To support learning [across the organization], it is important to find a sense of common purpose and focus.'

However, the case study showed that DollarCo was a long way from this unitary ideal. Instead of being able to rely on alignment around a common interest, management were learning about competing interests and priorities, most notably between management's economic interest in flexibility and productivity, and employees' interest in enhancing their positions or, at least, trying to protect against loss of status, resources and jobs.

### 7.2. Management Learning to Apply Pressure to Perform

When the first centralized DollarCo sites had been being established some years earlier, management tended to have an unsophisticated approach to employee relations. However, by the beginning of the case study, they had learnt a great deal about applying pressure to perform.

For example, they had learnt the value of establishing clear objectives and measuring progress against these. One reason management favored New PPM was that it incorporated significantly more opportunities to measure what staff do. According to one of the managers driving the change to New PPM: 'We are measuring them now on their work, but [will do even more] with New PPM. There could still be some resistance from people who always look busy, but now they will actually have to BE busy! It will tighten up a few loopholes.'

Management were also learning to refine the link between pay and performance. In the words of an internal consultant, New PPM represented an attempt to move people away from the mindset of 'I will sit here and get paid, just doing enough to get by' to 'effort equals favorable [pay and skill development opportunities]'.

### 7.3. Management Learning about Information Management to Achieve Sought-After Outcomes

Resonant with Lukes' second face of power, as the change at DollarCo progressed, management were learning to package things so they were more acceptable to employees. Thus, a member of the New PPM project group described the kind of discourse that typically accompanied presentations:

'Employees were saying "isn't it a bit unfair that the person that talks the loudest, or negotiates the strongest, ends up with a high salary?". But we were saying "hang on, we're trying to create a "free labor market" in here—owning and influencing your own career".'

The emphasis in these comments on 'free labor market' was part of a broader discursive agenda which recognized that, to achieve the kinds of performance that management were seeking, a new conceptual vocabulary would be needed. In the words of one manager associated with the change, New PPM was about 'fundamentally changing the way people view their work': '[New PPM] is all about work environment and culture. We talk about "team-based culture", "providing a vision" and "engaging and involving others".'

Learning how to present the changes so that they were more readily accepted was a subset of a larger body of learning relating to managing communication of information in the service of the company's economic and technical interests. Most notably, DollarCo management were learning how to depict New PPM as an attractive, rational and well-structured technical intervention. At the same time, they were learning to downplay aspects of the system which challenged employees' ontological interest—for example, to downplay the fact that they intended to increase the monitoring of each

employee's productivity level, and instead emphasize the increased opportunities for skill formation that accompanied New PPM.

They also learnt about the value of small group dialogue. At first, those driving New PPM relied on large group information sessions for staff, but this gradually changed:

> 'We learnt that to get the message across, you have to do it in small groups. We [initially] had communication forums...in groups of 40 to 60 people at a time. Everybody got the same information, but [later, senior managers started going] out to the team meetings, and talking to them about the issues.'

This comment reflects management learning about the value of presenting information to small groups with a common interest, something particularly important in their dealing with older Product Center employees, the group most resistant to change.

## 8. Examples of Shared Learning Associated with Employees Protecting against the Exercise of Power

Consistent with Habermas and the material summarized in Table 1, as DollarCo management were learning to exercise power in service of DollarCo's economic and technical interests, employees were learning to resist. In particular, they were learning to protect their own and their co-workers' interests and, in the process, learning more about 'money and power'. Examples are given below.

### 8.1. Employee Learning Associated with Protecting Co-Workers' Interests

During the period leading up to the vote on New PPM, there was considerable debate amongst employees about potential impacts on pay and performance management. According to a middle-level manager, 'There is a lot of baggage from previous roles, under prior structures. People believe their jobs have been downgraded.'

As this comment suggests, during DollarCo's turbulent history, groups of long-serving employees had acquired 'baggage' (that is, learning) from situations where feelings of self-worth and of being in control were threatened. Interests were at the heart of this learning. Just as management learning was often associated with DollarCo's economic and technical interests, employee learning from the past often related to their ontological interests, and this learning made them wary of New PPM. For example, employees expressed concern about the new pay and performance management system's 'sugar coating' and about it being a 'Trojan horse', and they speculated about the 'sting in its tail'. Product Center staff had also learnt from colleagues at a Center in another state that the changes could constitute loss of control over their work.

A great deal of employee learning at DollarCo also related to the larger economic and political contexts in which the Centers operated—in essence, to the ruthlessness of 'money and institutional power' (to quote Habermas (1987)) and its threat to the ontological interest. Some long-serving employees at DollarCo had learnt that senior management viewed them as just another expendable resource. Describing the company's inflamed employee relations history, a team leader explained:

> 'Some years back, we employed lots of people to replace the ones retrenched, and we've still got people who saw that strike. It was a mess. The majority left, and the ones who stayed keep saying "remember back then, when that happened?".'

This shared learning from the past greatly impacted on employees' receptiveness to New PPM. According to a DollarCo manager: 'I think that some people just won't change because of the damage that was done back then. They are in the stage of denial.' For the individuals involved, the phrase 'the stage of denial' suggests stages of grieving (Kubler-Ross 1969), reflecting the deep level of trauma experienced by some employees in the transition to centralized sites and the potential for them to lose status or jobs. As DollarCo had changed, some employees had learnt harsh lessons about the influence of money and institutional power on their work and about their inability to trust management, and had shared that learning with co-workers with common ontological interests.

*8.2. Employee Learning about 'Money and Power'*

Some DollarCo employees harbored deep-seated fears about the consequences of economic and political forces on their jobs and lives. A Product Center team leader observed:

> 'A hypothetical example might be "everyone has to start at 6:30 a.m. because we have to get an edge on the market and start before everyone else". Some people would think this is ridiculous. "Well then, if you don't come to the party, you might have to look at working elsewhere".'

Asked about whether senior management were getting better at creating performance management systems that made it difficult for people to object to doing whatever they were told, an employee replied: 'Yes, that's a big issue. But I don't believe that's the monster behind the concept—the beautiful part is creating a dynamic learning environment.'

Given nothing monstrous has been suggested by the interviewer, these comments seem to suggest that some employees feared a 'monstrous' ruthlessness lurking behind the 'beautiful' rhetoric. In this scenario, management would have absolute power over working conditions. Perhaps to reassure himself and the interviewer, no sooner is the existence of such a 'monstrous' scenario acknowledged than it is interrupted with the reference to the hoped-for 'beautiful' and 'dynamic' nature of New PPM.

In another interview, a team leader was asked about the attitudes of employees who opposed New PPM. He replied 'I have heard things like "this is silly, do I have to do this?". Every matter has to be explored individually. If someone is just around the corner from retirement, this could be a chop in the arm for them.' There can be little doubt that the phrase 'a chop in the arm' refers to a painful, violent assault, indicative of the level of confrontation to the ontological interest that New PPM represented. Well-paid, older Product Center employees felt particularly threatened, because they did not have skills to match the new context. There seemed to be no satisfactory pay solution for them. As one of the team leaders commented: 'They are afraid that they have reached the peak. Where can they go from there? To add insult to the wound [sic], if you then realize you have to change, it's another six months before you will be judged [against the competency framework].'

The terms used here—'afraid' and the reference to 'wound'—once again underline the vulnerability and distress felt by groups of long-serving employees who were learning that both their immediate work context and the broader industrial environment was being overturned by a new economic and political order.

## 9. Discussion

This paper has called for more attention to interests and interest differences in studies of organizational learning. Drawing on theoretical work by Habermas (1987) and Lukes (2005), it has argued that, in parallel with learning associated with resolving overt challenges relating to an organization's economic and technical interests, a great deal of learning occurs at a second, covert level as well (see Table 1). This second level encompasses learning associated with management's efforts to exercise power, and employees' learning from their efforts to protect their interests in the face of management power. In both cases, interests and interest differences play a central role. Management's exercise of power primarily serves the organization's economic and technical interests; and employees' attempts to protect against the exercise of power primarily serves their ontological interest.

The DollarCo data indicate that a great deal of shared learning was occurring as New PPM was being introduced. Consistent with Habermas, much of this learning was associated with tensions between DollarCo's (and their management representatives') economic and technical interests, and employees' ontological interests. Management were learning about pluralist aspects of work, about the most effective ways to pressure employees to perform and about using information to achieve sought-after outcomes. At the same time and largely in response, employees were learning to protect their own and their co-workers' interests and, in the process, were learning about 'money and power'.

The case study contained many examples of the lessons that resulted from these challenges being shared with others whose interests coincided.

Let us now consider requirements for this shared learning to become 'organizational'. The proposition that, in a commercial organization like DollarCo, learning relating to change in a pay and performance management system should be 'organizational' seems a reasonable one. After all, most large companies change their approach to pay and performance management from time to time, and this area is fundamental to the success of many organizations. One hopes that these changes are not random, but rather reflect the lessons of previous organizational learning about reward and recognition. Certainly, DollarCo managers and employees repeatedly stated that 'we learnt', with management often seeming to imply that 'we' was the whole of DollarCo. However, in most cases, it seemed to this investigator that 'we' was, in fact, a group with whom the interviewee shared common interests. Without taking interest differences into account, these comments might have been taken as confirmation that organizational learning was occurring.

However, by paying attention to interest differences, it seemed clear that learning about pay and performance management at DollarCo actually resided at the level of 'shared interest group' rather than 'organization'. Individual managers responsible for introducing New PPM were learning new things during the case study period, and often shared their learning with fellow managers. Similarly, employees were learning, and often shared their learning with co-workers, particularly across the Product Center where the threat to their ontological interest was greatest. There was no evidence, however, that lessons learnt about pay and performance management were ever elevated above the level of 'shared interest group learning'.

This finding raises questions about the circumstances under which 'shared interest group' learning becomes 'organizational' learning. The data reported in the last two sections, and the conceptual material that precedes it, points to a great deal of reluctance to share significant learning with others unless interests coincide.

Sometimes, economic, technical and/or ontological interests may extend across whole organizations. For example, a small knowledge-intensive company might have a remuneration system that rewards technical collaboration and knowledge sharing to such an extent that all organizational actors approximate a single shared technical-economic interest group, conceivably providing a basis for organizational learning across the company.

Or one might visualize an organization where a challenge to all members' ontological interest (e.g., the threat of site closure by a multinational company) could mean that the site approximates a single shared ontological interest group for a period. Such a situation could conceivably result in site-wide learning by managers AND employees about enterprise vulnerability in the face of international capitalism. In both examples, what could flow from such circumstances is consistent with definitions of organizational learning. That is, organizational members could learn new things, and that learning could be shared, retained in various forms of memory and subsequently applied widely throughout the organization in ways that impact on productivity and profit. Much more commonly, though, the material considered in this paper suggests that economic, technical and/or ontological interests tend not to extend across whole organizations. Moreover, because there is reluctance to share lessons learnt with others whose interests differ, the locus for shared learning at work tends to be 'interest group' rather than 'organization'.

The relationship between workplace learning and shared interests deserves far more research attention than it has received to date. If the tentative conclusions presented here are supported by additional research, the implications for organizational learning scholarship will be profound—namely, that much of the learning that has formed the basis for accounts of organizational learning in recent decades could be described far more accurately as 'shared interest group learning' rather than 'organizational learning'. Such a change would not only represent a major conceptual shift, but would have important practical implications as well, most notably a change in emphasis for companies and their consultants from 'how can we improve organizational learning?' to 'what can we do to

maximize the extent that proposed changes accommodate the economic, technical and ontological interests of ALL organizational members??'

**Conflicts of Interest:** The author declares no conflict of interest.

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
