# Peer review of "Interest Differences and Organizational Learning"

_admsci, doi:10.3390/admsci7030026_

Round 1
Reviewer 1 Report
I really like the idea of more strongly highlighting interests and interest differences in this context. I also think that referring to Habermas and Lukes is a nice idea for this purpose.
What I miss in the current version of the paper, however, is a more extensive integration of the authors’ criticism and their theorizing into existing literature.
For example, the role of shared and competing interests has been taken into account in the context of knowledge management. Relevant keywords are “information-exchange dilemma” or “the social psychology of knowledge management”. The authors should definitely refer to these conceptualizations.
Moreover, a key issue in the paper is the “interplay between individual and collective knowledge” and there is also literature that deals with this issue, both in the context of organizational learning and in other contexts.
What is not clear in the paper is whether the authors deal with declarative or non-declarative knowledge. They should take these cognitive foundations of organizational learning into account (e.g., Kump’s paper in Frontiers in Psych, 2015).
There are other paragraphs that would benefit from references (e.g., lines 105-110).
I also think that the authors should clarify their goals more explicitly at the beginning of the manuscript. Currently, the purpose of the paper becomes clear to the reader only bit by bit. The very last paragraph of the first section, for example, should be placed (in a revised way, of course) at the beginning.
One point that I’m not really convinced of is the consideration of the cultural aspects in terms of Hofstede (page 3). I’m not sure whether these considerations are really required. I think it’s rather confusing than helpful. The authors should consider dropping this part.
Finally, I understand that the authors are very critical regarding the concept of organizational learning, but it might still be a good idea not to write this term in inverted commas every single time.
Author Response
Grammar and spelling have been re-checked and refined.
The introduction has been extensively revised as detailed in the ‘comments and suggestions for authors’ below. As part of these changes, there are now strong connections between the study and the related fields of knowledge management and workplace learning.
The relationship between results and conclusions has been strengthened, and the whole focus of the paper sharpened for clarity and consistency.
The authors of this manuscript aim at considering workplace learning through the lens of “interests” that play a role in this process.
I really like the idea of more strongly highlighting interests and interest differences in this context. I also think that referring to Habermas and Lukes is a nice idea for this purpose.
What I miss in the current version of the paper, however, is a more extensive integration of the authors’ criticism and their theorizing into existing literature. For example, the role of shared and competing interests has been taken into account in the context of knowledge management. Relevant keywords are “information-exchange dilemma” or “the social psychology of knowledge management”. The authors should definitely refer to these conceptualizations.
Accepted and rectified. I have now referred in some detail to these conceptualizations, in the process acknowledging that some attention has been paid to shared and competing interests in the field of knowledge management.
| Lines 73-113 |
Moreover, a key issue in the paper is the “interplay between individual and collective knowledge” and there is also literature that deals with this issue, both in the context of organizational learning and in other contexts.
Accepted and rectified. I have elaborated on this area, in the process drawing on the work of key contributors to individual / collective knowledge and learning discussions.
| Lines 79-89; 93-113; 148-50 |
What is not clear in the paper is whether the authors deal with declarative or non-declarative knowledge. They should take these cognitive foundations of organizational learning into account (e.g., Kump’s paper in Frontiers in Psych, 2015).
After carefully reading material that argues for a distinction between ‘declarative’ and ‘non-declarative knowledge’, including Kump’s paper, I have responded by using the declarative / non-declarative distinction to illustrate the tendency (one which is my paper’s central theme) in discussions of organizational learning to gloss over interests and interest differences. For example, in Kump’s paper, interests are not referred to; there is no mention of power or politics; and the emotional aspects of collective learning are explicitly ruled out (p. 9).
| Lines 57-72 |
There are other paragraphs that would benefit from references (e.g., lines 105-110).
Accepted and rectified. Here and elsewhere, the relevant references have been added.
| Lines 175-7 |
I also think that the authors should clarify their goals more explicitly at the beginning of the manuscript. Currently, the purpose of the paper becomes clear to the reader only bit by bit. The very last paragraph of the first section, for example, should be placed (in a revised way, of course) at the beginning.
Accepted and rectified. The introductory material has been reordered, as suggested. More generally, the central themes have been refined for clarity and consistency throughout.
| Lines 25-34 |
One point that I’m not really convinced of is the consideration of the cultural aspects in terms of Hofstede (page 3). I’m not sure whether these considerations are really required. I think it’s rather confusing than helpful. The authors should consider dropping this part.
Accepted and rectified by deleting this material.
|
Finally, I understand that the authors are very critical regarding the concept of organizational learning, but it might still be a good idea not to write this term in inverted commas every single time.
Accepted and rectified. In references to organizational learning, inverted commas are now used sparingly.
|
Reviewer 2 Report
Concise paper with a pertinent and very precise objective. The "9. Discussion" is very clear. The number of References is sufficient, although not very high.
I suggest rewriting the following text: "While the case study does not contain any examples of learning which could unequivocally be considered ‘organizational’, . . .".
Author Response
Grammar and spelling have been re-checked and refined.
Concise paper with a pertinent and very precise objective. The "9. Discussion" is very clear. The number of References is sufficient although not very high.
Accepted and rectified. To accommodate feedback from my two reviewers, there are now 50 references in place of the 25 contained in the first draft.
|
I suggest rewriting the following text: "While the case study does not contain any examples of learning which could unequivocally be considered ‘organizational’, . . .".
Accepted and rectified. This passage has been removed, and the emphasis on shared interest group learning, and the particular circumstances in which ‘organizational’ learning might occur, have been clarified.
|
Round 2
Reviewer 1 Report
This is a successful revision.
I don't see the need for further modifications.